# Crystal Structure and Mechanical Properties of Uniaxially Stretched PA612/SiO_2_ Films

**DOI:** 10.3390/polym12030711

**Published:** 2020-03-23

**Authors:** Yichao Wu, Anmin Huang, Shuhong Fan, Yuejun Liu, Xiaochao Liu

**Affiliations:** 1Key Laboratory of Advanced Packaging Materials and Technology of Hunan Province, Hunan University of Technology, Zhuzhou, Hunan 412008, China; cliffwuther@163.com (Y.W.); f1s1h1@163.com (S.F.); 2Zhuzhou Times New Material Technology Co. Ltd., Hunan Zhuzhou 412008, China; huanganmin@csrzic.com

**Keywords:** crystal structure, mechanical properties, uniaxial stretching, polyamide 612

## Abstract

Stretching has a significant effect on the microstructure and ultimate performance of semi-crystalline polymers. To investigate the effect of stretching on structure and mechanical properties of uniaxial stretched PA612/SiO_2_, PA612 and PA612/SiO2 films were prepared at four temperatures close to the glass transition temperature at various strain. The samples were characterized by a transmission electron microscope (TEM), wide-angle X-ray diffractometer (WAXD), Two-dimensional wide-angle X-ray Scattering (2D-WAXS), differential scanning calorimeter (DSC), dynamic mechanical analyzer (DMA), and stretching tests. The results showed that the α phase was the dominant phase in PA612 casting film, no obvious γ phase was observed, while both stretching and the presence of SiO_2_ can induce the generation of α phase and improve the crystallinity of PA612. Crystals were oriented along the stretching direction and the b axis was parallel to the equatorial direction after stretching. The interplanar spacing of (010/110) decreased with the increasing stretching temperature and expanded with the increasing strain, while stretching temperature and strain present negligible effect on the interplanar spacing of (100). The grain size increased with the stretching temperature while decreased with strain. The presence of SiO_2_ led to reduce the yield stress and the stress drop beyond yielding of the composite. Uniaxial stretching gave rise to a significant improvement in the fracture stress and the glass transition temperature.

## 1. Introduction

Due to the exploitation of petroleum, the development and application of biobased or semi-biobased polymers which are ideal substitutes of petroleum-based polymers have been a hot research topic. Nylon 612 (PA612) is a semi-biobased material with excellent comprehensive performance. Its raw material, dodecanedioic acid, can be obtained from vegetable oil derivatives and the biobased content in Nylon 612 can reach up to 66% [1]. Meanwhile, PA612 exhibits advantages in low moisture sensitivity, high flexibility, and ductility over short-chain nylons (e.g., PA6, PA46) and superior mechanical strength compared with long-chain nylons (e.g., PA1012, PA1212). Therefore, PA612 can be applied in various fields, including automobile manufacturing, aerospace engineering and electronic appliances [2,3]. PA612 films exhibits great potential as a packaging material in virtue of its low moisture absorption, good dimensional stability, and high mechanical strength. 

The ultimate properties of polymers is governed by the microstructure such as crystal structure, crystal orientation and crystallinity, which is significantly affected by external conditions, such as low-intensity γ radiation, temperature gradient, and stretching field. During γ radiation, a part of molecular chains was broken which impassion that low molecular weight polymers were generated resulting in reorientation of molecular chains in the amorphous zone and increased crystallinity [4,5,6]. Temperature slope crystallization is another way of inducing the orientation of the molecular chain, in which the melt is induced to crystallize in the lower temperature direction by the temperature difference between both sides of the sample [7]. The crystal orientation and conditions for positive and negative spherulites in PA612 during temperature slope crystallization had been investigated by Yoshida [8]. To apply a stress field is an effective approach to induce crystal orientation and crystal transition during stretching. Wang reported a thorough study of the crystal structure of PA612/PA1012 composite and its crystal transition during stretching [9,10,11]. The Brill transition of α crystal of PA1012 in composite to γ crystal was observed during heating, while stretching was mainly responsible for the induction of γ crystal of PA1012. However, the crystal transition of PA612 was not observed during heating due to the Brill transition temperature of PA612 is close to its melting temperature [12].

Stretching not only affects crystallization and crystal orientation of nylons, but also their macroscopic properties [13,14,15]. Liu prepared uniaxial and biaxial stretched PA6-66/MMT composite films [16]. After stretching, the crystallinity of composite films was enhanced and the transition of β phase to α phase was observed. The mechanical strength of composite films is highly improved and it is mainly affected by the stretching ratio. Uniaxial and biaxial stretching led to decreased transmittance and increased haze in the PA6-66/MMT films and the thermal shrinkage of the films increased with increasing stretching ratio.

Stretching is commonly observed in film fabrication process (e.g., blowing, casting, and biaxial stretching). During stretching process the molecular chains of semi-crystalline polymers prefer to orient along the stretching direction and the transition of spherulites to highly oriented shish-kebab structure was observed [10]. In this case, the crystallinity of semi-crystalline polymers increases, and the stretching-induced crystal orientation [17] and crystal transition [18,19], which are two key factors that remarkably affect the mechanical performance of materials [20]. Hitherto, the reports regarding the crystallization behavior of PA612 is rather limited. Moreover, to the best of our knowledge, the stretching-induced crystallization, and the structure-property relationships of PA612 films has not been systematically reported, which could be a sever hindrance for the application of PA612 films. 

The modification with nanoparticle is an effective approach to improve the ultimate properties of polymers. In virtue of their small sizes and high specific areas, nanoparticles can be used as nucleators to induce crystallization of polymers and enhancing their performance. For instance, the amino-modified Al_2_O_3_ flakes have been convinced to effectively improve the strength, abrasion, and toughness of PA612. The high content nano-TiO_2_ can induce the formation of γ crystals and increase the melting point of PA612 [21]. Nanoscale SiO_2_ is also an important particle to significantly enhance the tensile strength and impact strength of polymers. Serving as a nucleator, SiO_2_ can enhance the stretching strength and storage modulus, and increase decomposition temperature and glass transition temperature [22]. The recent reports demonstrated that SiO_2_ induced the generation of γ crystals, thus enhancing the crystallinity of PA6 and reducing the grain size. It was also observed that by addition of less than 3 wt.% SiO_2_, the impact and bending strength of PA6 can be simultaneously improved [23,24]. To date, the incorporation of SiO_2_ into PA612 matrix has not been investigated, which is yet significant for the enhancement of macro-properties of PA612.

In this work, nano-SiO_2_ was incorporated into PA612 by melt blending to fabricate nanocomposites, and the PA612 nanocomposites were melt casted into films, followed by uniaxial stretching at various parameters. The effects of SiO_2_ and the stretching process on the morphology, crystallization behavior, the crystal orientation of PA612 films were investigated in detail, and furthermore the mechanical properties of the stretched films were studied to shed light on the processing-structure-property relationships of the PA612 nanocomposites.

## 2. Materials and Methods 

### 2.1. Materials 

PA612 (Zytel 151L NC010) was purchased from Dupont Co. (DE, USA). Nano-SiO_2_ modified by KH550 with the particle size of 30 nm was supplied by HongSheng Co. (Zhejiang, China). 

### 2.2. Preparation of the PA612/SiO_2_ Composite and Film

SiO_2_ and PA612 were dried in a vacuum oven at 80 °C for 4 h, after which the composite were pre-mixed using a high-speed mixer for 5 min. The PA612/SiO_2_ composites were prepared via melting compounding using a twin-screw extruder with a screw speed of 165 rpm. The temperature profile ranges from 210 °C to 240 °C. The neat PA612 experienced the same extrusion procedure for comparison. Then the films were prepared by a casting extruder with the temperature profile ranges from 240 °C to 260 °C, and the thickness of the film was controlled to be ca. 0.12 mm. Composite films with different SiO_2_ content (0 wt.%, 3 wt.%) are assigned as S0 and S3, respectively.

### 2.3. Preparation of Uniaxially Stretched Films

According to ISO 527, the casting films were cut into dumbbell-shaped samples with the length of 120 mm and width of 6 mm using a punching machine. The films were then uniaxially stretched by a universal testing machine equipped with a heater. During the experiment, different stretching ratios were selected to perform the stretching process at four temperatures of 30 °C, 80 °C, 120 °C, and 160 °C. The films were stretched at a speed of 50 mm/min after preheated for 5 min in stretching temperature and then quenched to room temperature.

### 2.4. Transmission Electron Microscopy (TEM) 

PA612/SiO_2_ composite films were sliced with an ultra-thin microtome after embedded with epoxy resin. The sheet was placed on a copper grid for observing on JEOL 1230 (SANYO, Osaka, Japan) with a voltage of 90 KV.

### 2.5. Differential Scanning Calorimetry (DSC) 

Q20 instrument (TA Instruments Co, Delaware, USA) was employed to carry out the thermal analysis. The sample was heated from 25 °C to 250 °C under a nitrogen atmosphere with the heating rate of 10 °C/min. The crystallinity (Xc) of the sample is calculated by the following formula:(1)$$\mathrm{Xc}=\Delta H_{f}/\left[(1-\varphi )\cdot \Delta H_{f}^{0}\right]$$
where Xc is the crystallinity, φ is the content of SiO_2_, Δ$H_{f}$ is the melting enthalpy, Δ$H_{f}^{0}$ is the melting enthalpy when PA612 is completely crystallized, Δ$H_{f}^{0}$ of PA612 is 258 J/g according to the report [21].

### 2.6. Wide-Angle X-Ray Diffraction (WAXD)

WAXD measurements were carried on the Rigaku Ultimate IV X-ray diffractometer (Rigaku, Tokyo, Japan) with Cu Kα radiation (λ = 0.154 nm). The tests were under reflection mode with a range of 5°–40° and a scanning rate of 4°/min. The interplanar spacing d and the grain size L_hlk_ are calculated by Equations (2) and (3), respectively:d = λ/2sinθ(2)
L_hlk_ = (K_α_·λ)/(β·cosθ)(3)
where d is the interplanar spacing, λ is the wavelength of the Cu Kα radiation (0.154 nm), θ is half of the diffraction angle, L_hlk_ is the average grain size perpendicular to the hlk crystal plane, Kα is the grain size constant (0.89), β is the full width at half maximum.

### 2.7. Two-Dimensional Wide-Angle X-Ray Scattering (2D-WAXS)

2D-WAXS measurements were carried out on a Xeuss 2.0 system (Xenocs SA, Sassenage, France) with Cu Kα radiation. The detector was set perpendicular to the film surface with a distance of 183.71 mm and an exposure time of 5 min. All 2D WAXS patterns were corrected by the Fraser method to compensate the detection of a flat-plat detector.

### 2.8. Dynamic Mechanical Thermal Analysis (DMA)

The dynamic mechanical analysis was conducted using a DMA242C dynamic mechanical analyzer (TA Instruments Co, Delaware, USA) at a frequency of 2 Hz with a heating rate of 3 °C/min from −20 °C to 120 °C in the tensile mode. The glass transition temperature (T_g_) was determined from the peak of loss factor (tanδ) vs. temperature curves.

### 2.9. Tensile Testing

The uniaxially stretched films were tested by an electromechanical universal testing machine (Wance testing machine Co, Shenzhen, China) in the tensile direction at room temperature of 25 °C with a tensile rate of 50 mm/min.

## 3. Results

### 3.1. Morphology of SiO_2_ in Films

The TEM graphs of PA612/SiO_2_ composite films with different stretching ratios are shown in Figure 1. It can be clearly observed that the SiO_2_ particles are liable to agglomerate in the PA612 matrix. Notably, compared with the unstretched films (ε = 0), the agglomerated SiO_2_ particles in the films with strain of 2 begin to move along the stretching direction due to the stretching-induced orientation of the molecular chains. As a result, the SiO_2_ particles slip and transform into a beaded morphology.

### 3.2. Crystallization Behavior of Uniaxial Stretched PA612 and PA612/SiO_2_ Films

Figure 2 illustrates the primary heating DSC curve of PA612 before and after stretching, and the corresponding data are collected in Table 1. As shown in Figure 2a, a wide broad recrystallization peak in 190–210 °C was found for each unstretched films, which could be attributed to the incomplete crystallization of nylon crystals due to the films quenching on the cold chill roll during film casting process. In addition, this peak is more intense for the film annealed at high temperature. The crystallization of nylon films was facilitated and the crystal lamellae were thickened during high temperature annealing [25]. However, this peak is absent for the stretched films. This could be attributed to the strain-induced crystallization of the films. Additionally, the crystallinity of PA612 was reduced after heat treatment at different temperatures, which is consistent with PA66, it may be attributed to the rupture of ordered structures caused by the recombination of molecular chains [26]. As shown in Figure 2b, the melting point of PA612 raises as the increasing of stretching temperatures. This may be attributed to the fact that high temperatures favor the movements of molecular chains. As a result, crystal growth rate and crystallization degree are higher than those in the lower temperature.

The DSC curves of uniaxially stretched PA612 and PA612/SiO_2_ films at the stretching temperature of 120 °C under varied strain are shown in Figure 3, and the corresponding data are listed in Table 2. Two melting temperatures (ca. 205 °C and 218 °C) are found for the films stretched at the strain ε = 2 and ε = 2.5, regardless of the addition of SiO_2_ particles. In addition, the melting enthalpy of neat PA612 films at 205 °C is lower than their PA612/SiO_2_ counterparts. Zhang et al. demonstrated that the crystals with deficient regularity are feasible to melt in the lower temperature, and the high melting temperature is attributed to the melting of perfectly crystallized spherulites [27]. In the DSC curves of the films stretched at high strain, the presence of low temperature melting peaks suggests the generation of imperfect crystal fragments, which should be originated from the stretching-induced extraction of lamellae from spherulites [28]. For PA612 films, the low temperature melting enthalpy increased from 1.02 J/g at ε = 2 to 6.71 J/g at ε = 2.5, indicating that the amount of crystal fragments is positively related to the strain. This phenomenon is more significant for PA612/SiO2 composite films, in which the low temperature melting enthalpy of PA612/SiO_2_ composite increases to 17.29 J/g at ε = 2.5. It revealed that the SiO_2_ particles increase the crystallinity of the composite films and affect the movement of crystal lamellae during stretching.

It can also be found from Table 2 that the crystallinity ascends with the strain regardless of the incorporation of the SiO_2_. It indicates that the presence of strain-induced crystallization of molecular chains for the stretched films. It should be noted the addition of SiO_2_ is capable of improving the crystallinity for both unstretched and stretched films. Specifically, the crystallinity of PA612 films enhance with the increasing of strain, and the maximum value appears at ε = 2 (34%). However, as the strain is higher than 2, the crystallinity decreases. A similar tendency can be found for PA612/SiO_2_ films at low strain, whereas the crystallinity of PA612/SiO_2_ films increased at ε = 2.5 dramatically. The reduction of the crystallinity of PA612 films at ε = 2.5 may be attributed to the over-high stretching ratio, giving rise to the rupture of the crystal spherulites [29,30].

To further investigate the variation of crystal structure during stretching, the films were characterized by WAXD, as shown in Figure 4 and Figure 5. Two diffraction peaks are observed at approximately 20° and 23° in each curve, corresponding to the (100) and (010/110) plane of α crystals respectively. It indicates that only α phase crystals exist in the PA612 or PA612/SiO_2_ films after stretching, since the characteristic diffraction peak of γ phase occurs at 21° [31]. It indicates that the α phase is the dominant phase in PA612, which is different from the casted PA6 and PA1012 films that mainly contain γ phase crystals [9,32]. As shown in Figure 5b, the diffraction peak of (100) planes of unstretched PA612 film is more intense than the (010/110) planes, suggesting that the PA612 crystals prefer to grow along the direction perpendicular to (100) planes during quenching in the film casting procedure [33]. The (100) planes of PA612 are perpendicular to the hydrogen bond planes, the molecular chains move rapidly along the film casting direction with low activation energy and high steady nucleation free energy, which favors crystal growth [8]. Notably, the intensity of diffraction peak of (100) planes in the unstretched films (ε = 1) is much lower than the stretched counterparts. This should be ascribe to the twisting and rotation of crystal lamellae caused by the stretching [34]. As shown in Figure 6, the c axes of α phase, which is parallel to the direction of molecular chains, show preference orientation along the MD after stretching. The (010) planes present the priority of orientation during stretching, resulting in a rapid increment of diffraction peak intensity after drawing [35].

The 2D-WAXS patterns (Figure 6) illustrate the crystal orientation of the PA612/SiO_2_ films. In the 2D-WAXS patterns of unstretched samples, the two homogeneous Debye-Scherrer diffraction rings are observed at ca. 20° and 23°, corresponding to (100) and (010/110) planes of α crystals, respectively [25,33]. The diffraction signals of the two diffraction rings along the meridian disappeared and concentrate along the equator after stretching, which indicates the orientation of the crystal because of the increased strain. Moreover, the films with higher strain present reduced the arcs more, and the (010/110) planes show more concentrated intensity than (100) planes. It shows that the orientation degree of the crystal planes along the meridian direction increased with the strain and the orientation rate of (010/110) planes is higher than that of (100) planes.

The interplanar spacing (d-spacing) and grain size of the stretched films as a function of stretching temperature and stress are illustrated in Figure 7 and Figure 8, respectively. The d-spacing of (010/110) plane in stretched PA612 and PA612/SiO_2_ films decreases with increasing of stretching temperature. This can be accredited to the slip, deflection, and recombination of crystal lamellae during stretching at high temperature [36,37]. As shown in Figure 8, compared with the neat PA612 films, the PA612/SiO_2_ films exhibit an analogous tendency in the d-spacing of the two diffraction planes and the grain size of the spherulites. It implies that the addition of SiO_2_ plays an insignificant role in the crystal phase and structure transition of PA612 during stretching. The grain size of PA612 was apparently affected by stretching temperature and strain, and it is climbing with the stretching temperature increases. Hence, high stretching temperatures conduce to complete the grains during stretching. After stretching, the grain size declined drastically and remained constant after ε = 1.5. The addition of SiO_2_ led to a minished the grain size, indicating the grain refining effects of SiO_2_.

### 3.3. Mechanical Properties of Uniaxial Stretched PA612/SiO_2_ Films

Figure 9 demonstrates the stress-strain curves of S0 and S3 at different temperatures. It can be found that the stretching temperature and the presence of SiO_2_ both affect the yield behavior of the films. As increasing of stretching temperature, the yield stress gradually decreases and the yield point disappears as the temperature is higher than 80 °C. The presence of SiO_2_ leads to the diminution in the yield stress, as well as reduces the stress drop (Figure 9b) at the stretching temperature of 30 °C. Yalcin et al. revealed that the necking of the samples is proportional to the stress drop [33]. Hence, the incorporation of SiO_2_ can alleviate film necking during stretching. Meanwhile, the samples stretched at 30 °C exhibit a typical stress-strain behavior of typical semi-crystalline polymers. It consists of elastic region, yielding, strain-softening, necking, and orientation hardening stages. Nevertheless, as the stretching temperature exceeded 80 °C, the yielding and strain-softening region disappear and the curve transform from the elastic region to the orientation hardening directly without necking stage. Generally, the nonreactive SiO_2_ serves as a plasticizer within the matrix, while the reactive SiO_2_ present a reinforcement effect or a plasticizing effect due to its processing method and the reactivity of the grafted groups on SiO_2_ surface [22,23,38]. As shown in Figure 9, the yielding point of Sample S3 was lower than that of Sample S0, and it is difficult to explain why the yield stress of S3 samples are lower than S0 samples. On the one hand, it couldn’t attribute to the plasticizing effect under these conditions, because the peak of loss factor of S0 is higher than S3 and the glass transition temperature of S0 is lower than S3 (Figures 11 and 12). On the other hand, it is not the reason for the changing of crystal structure; the SiO_2_ particles diminished the grain size (Figure 8), and the yield stress of S3 should be higher than S0. This phenomenon should be investigated further. As the stretching temperature increased, the stress-strain curves of S0 and S3 tend to overlap each other, indicating that the effect of SiO_2_ on the matrix is negatively related to the mechanical properties at high stretching temperature. 

The stress-strain curves of the stretched PA612 and PA612/SiO_2_ films with different strains are shown in Figure 10. Herein, all samples were stretched at room temperature (25 °C). The stretched PA612 films still show yielding and strain-softening at the ε = 0 and ε = 1, but disappeared when the films were stretched to high strain and the stretched PA612/SiO_2_ samples exhibit no significant yielding or strain-softening region. Combined with Figure 6 and Figure 10, it can be concluded that the fracture strength of the films after stretching are positively correlated with the orientation of the crystal. Moreover, as the stretching strain increasing, the crystal orientation turn more obvious, the elongation at break decreases and the fracturing strength increases dramatically, and this is related to the orientation degree of crystal and the reorientation of polymer chains. This could be due to the development of fibrous crystals at high strain after the dislocation, deflection, and orientation of the crystals [39]. The movement of the fibrous crystals and the amorphous region is strongly hindered after sufficient stretching with high strain. The molecular chains in the amorphous region were highly oriented. The followed forced stretching would bring about the extraction and straightening of molecular chains, and the completed crystals are damaged into smaller crystallization units under stress [36]. A part of the molecular chains were extremely stretched before fracturing [40,41]. As a result, for the PA612 samples, the fracture strength increases and the elongation at break decreases as increasing of stretching strain, where the maximum fracture stress of stretched films was three times than that of unstretched films and the elongation at break is reduced from 160% to 35%. These changes can be attributed to the hindered movement of molecular chains in the amorphous regions and highly oriented fibrous crystals [35]. The PA612/SiO_2_ samples exhibit similar trends of fracture stress and elongation at break, but they have higher fracture stress and lower elongation at break. This is because the SiO_2_ hinders the movement of molecular chains when the films were stretched. 

The storage modulus and tanδ as a function of temperature curves of uniaxially stretched PA612 and PA612/SiO_2_ films are shown in Figure 11 and Figure 12, respectively, from which we can obtain the glass transition temperature (Table 3) and investigate the toughness-rigidity transition of the films. Indeed, the glass transition temperatures were highly improved. As increasing of stretching strain, the storage modulus increases accompany with the decreases of the loss factor peak and the monotonically elevated glass transition temperature. It indicates that the increased strain would lead to enhanced rigidity and reduced toughness. Moreover, the glass transition temperature and storage modulus of PA612/SiO_2_ films were higher than that of PA612 films regardless of the stretched or unstretched films and this is consistent with PA6/SiO_2_ composites where the SiO_2_ particles improved the glass transition temperature and storage modulus of PA6 [38]. In the PA612 samples, the tanδ vs. temperature curves show two peaks when the ε = 2 and ε = 2.5 from Figure 11b, which would help to confirm that the molecular chains were extracted from the stretching-induced lamellae fragments from spherulites. However, this phenomenon was not shown in stretched PA612/SiO_2_ films.

## 4. Conclusions

In this study, uniaxially stretched PA612 and PA612/SiO_2_ films were prepared with the electromechanical universal testing machine; the crystal structure and mechanical properties were investigated in detail. The PA612 casting films were dominated by the α phase and this was not affected by stretching temperature or strain. The crystal was oriented in the stretching direction and the b axis was paralleled with equatorial direction. The stretching temperature and strain had a positive and negative effect on the interplanar spacing of (010), respectively. Strain induced the generation of α crystals and effectively improved the crystallinity of PA612. Additionally, the addition of SiO_2_ did not affect crystal form and crystal structure of PA612, but led to refined grain size and affects the extraction of crystal lamellar. The addition of SiO_2_ led to reduced yield stress and the strain-softening beyond yielding of the composite, necking was relieved during stretching. After stretching, the loss factor peak of films decreased and the glass transition temperature increased from 45.2 °C to 81.4 °C for the S0 samples and 74.0 to 84.2 for the S3 samples. The maximum fracture stress of stretched film was three times of that of unstretched films, the elongation at break of stretched films dropped to 35% for S0 samples, and the PA612/SiO_2_ samples exhibit similar trends.

## Figures and Tables

**Figure 1 polymers-12-00711-f001:**
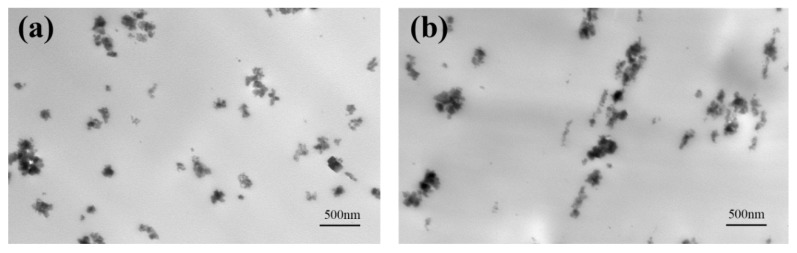
TEM graphs of the PA612/SiO_2_ films at the strain of (**a**) ε = 0 and (**b**) ε = 2.

**Figure 2 polymers-12-00711-f002:**
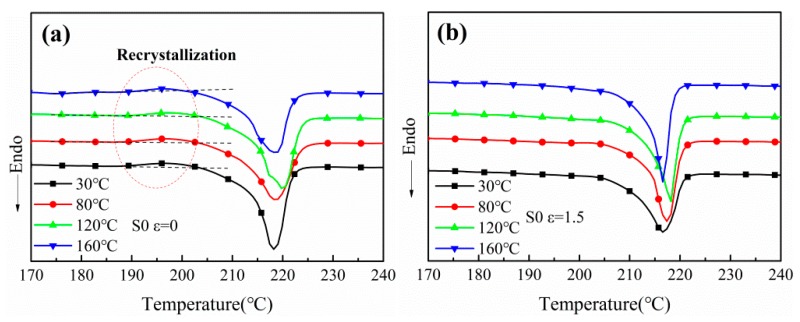
The DSC curves of PA612 films at the strain of (**a**) ε = 0 and (**b**) ε = 1.5.

**Figure 3 polymers-12-00711-f003:**
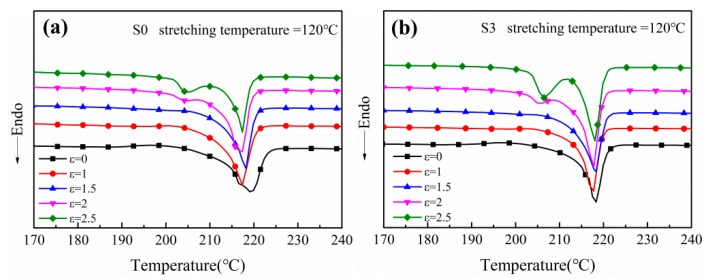
The DSC curves of uniaxially stretched (**a**) PA612 films and (**b**) PA612/SiO_2_ films at different strain.

**Figure 4 polymers-12-00711-f004:**
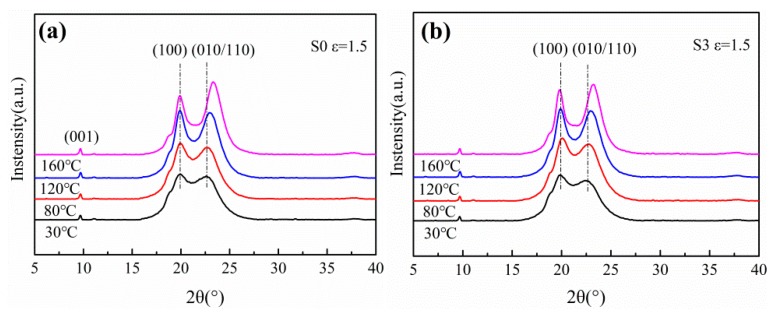
The WAXD curves of (**a**) PA612 and (**b**) PA612/SiO_2_ films uniaxially stretched at different temperature.

**Figure 5 polymers-12-00711-f005:**
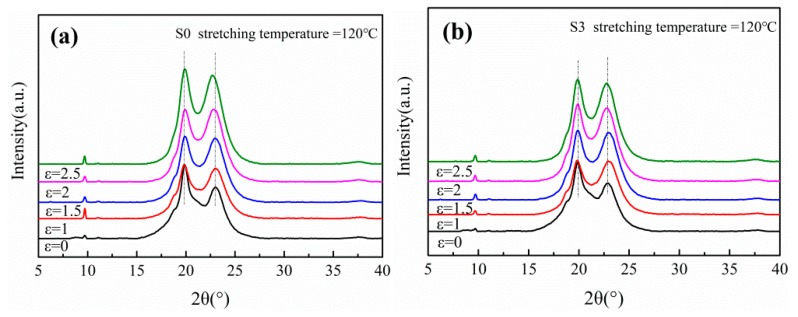
The WAXD curves of uniaxially stretched (**a**) PA612 and (**b**) PA612/SiO_2_ films at different strain.

**Figure 6 polymers-12-00711-f006:**
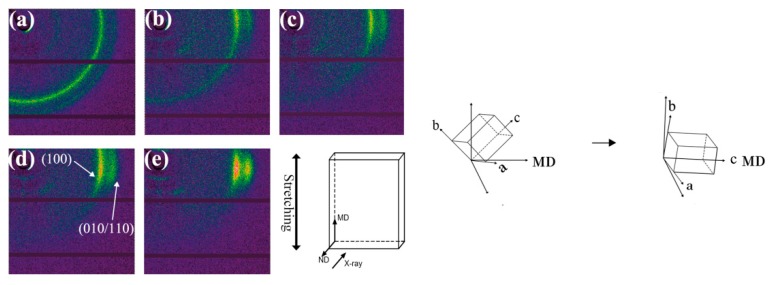
The 2D-WAXS of uniaxially stretched PA612/SiO_2_ films at the strain of (**a**) ε = 0, (**b**) ε = 1, (**c**) ε = 1.5, (**d**) ε = 2, (**e**) ε = 2.5.

**Figure 7 polymers-12-00711-f007:**
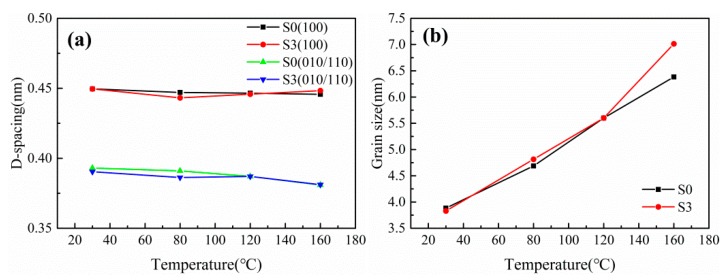
The (**a**) d-spacing and (**b**) grain size of uniaxially stretched PA612 and PA612/SiO_2_ films (ε = 1.5) at different stretching temperature.

**Figure 8 polymers-12-00711-f008:**
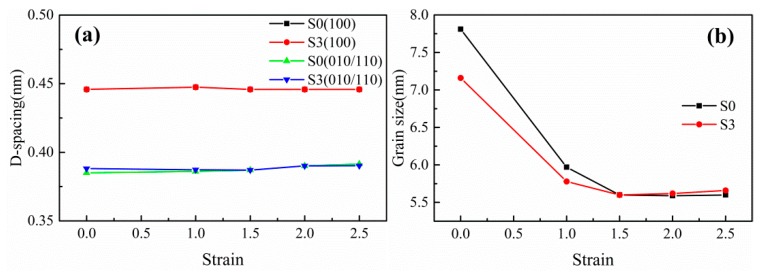
The (**a**) d-spacing and (**b**) grain size of uniaxially stretched PA612 and PA612/SiO_2_ films at different strain.

**Figure 9 polymers-12-00711-f009:**
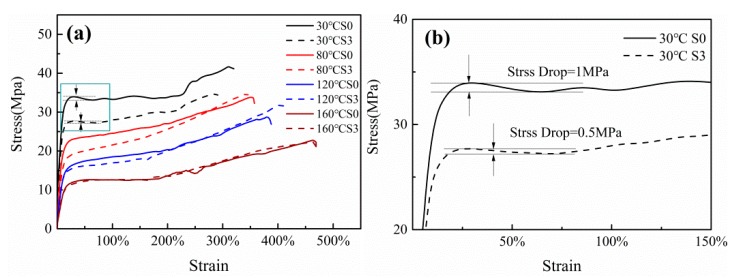
(**a**) the stress-strain curves of PA612 (solid line) and PA612/SO_2_ (dotted line) films at different temperature and (**b**) the zoom-in view of the curves at 30 °C.

**Figure 10 polymers-12-00711-f010:**
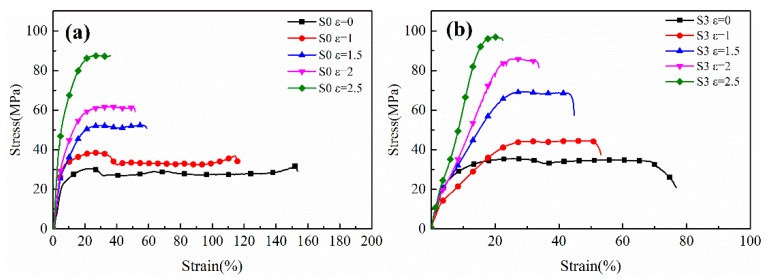
The stress-strain curves of the stretched (**a**) PA612 and (**b**) PA612/SiO_2_ films. (The curves were recorded at room temperature).

**Figure 11 polymers-12-00711-f011:**
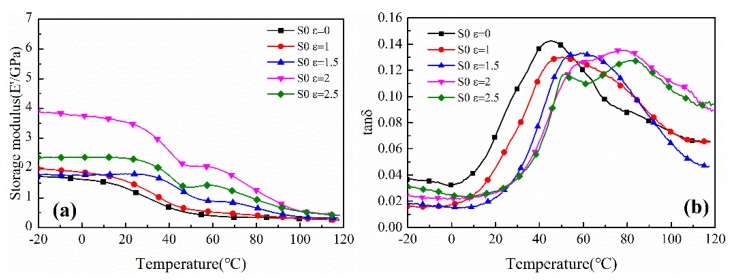
The (**a**) storage modulus vs. temperature and (**b**) tanδ vs. temperature curves of uniaxially stretched PA612 films.

**Figure 12 polymers-12-00711-f012:**
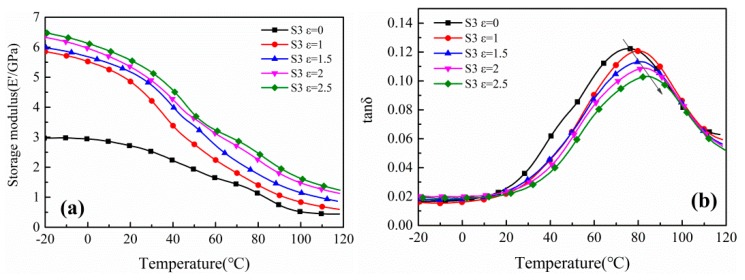
The (**a**) storage modulus vs. temperature and (**b**) tanδ vs. temperature curves of uniaxially stretched PA612/SiO_2_ films.

**Table 1 polymers-12-00711-t001:** The crystallinity for uniaxially stretched PA612 film.

Strain	Stretching Temperature (°C)	Melting Temperature (°C)	Melting Enthalpy (J/g)	Crystallinity (%)
0	30	218.2	68.47	26.5
0	80	218.6	62.25	24.12
0	120	219.1	63.69	24.69
0	160	218.6	62.79	24.34
1.5	30	216.8	65.12	25.24
1.5	80	217.5	67.97	26.34
1.5	120	218.3	70.54	27.34
1.5	160	216.5	71.01	27.52

**Table 2 polymers-12-00711-t002:** The crystallinity of uniaxially stretched PA612 and PA612/SiO_2_ films.

Sample	Strain	Melting Temperature (°C)	Melting Enthalpy (J/g)	Low Temperature Melting Enthalpy (J/g)	Crystallinity (%)
S0	0	219.1	63.69	_	24.69
S0	1	217.3	66.64	_	25.8
S0	1.5	219.8	70.54	_	27.34
S0	2	217.3	87.88	1.02	34.06
S0	2.5	217.3	72.48	6.71	28.09
S3	0	218.3	64.31	_	25.60
S3	1	217.7	66.36	_	26.52
S3	1.5	218.3	75.34	_	30.10
S3	2	217.7	83.18	3.68	33.2
S3	2.5	218.1	90.11	17.29	36.01

**Table 3 polymers-12-00711-t003:** The glass transition temperature of uniaxially stretched PA612/SiO_2_ films.

Strain	ε = 0	ε = 1	ε = 1.5	ε = 2	ε = 2.5
S0 (T_g_/°C)	45.2	50.5	58.6	62.1/79.35	50.2/81.4
S3 (T_g_/°C)	74.0	79.0	81.0	82.9	84.2

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
