# Peer review of "Crystal Structure and Mechanical Properties of Uniaxially Stretched PA612/SiO2 Films"

_polymers, 2020, doi:10.3390/polym12030711_

Round 1
Reviewer 1 Report
I can not understand the originality and characteristics of the PA612/SiO2 composites exhibited in this manuscript. The authors should add the experimental results and explanations to clarify the role of the SiO2 in the composites by considering the following comments.
1)
To discuss the role of the SiO2 on the mechanical properties of the PA612, the authors should add the stress-strain curves and the DMA results of the neat PA612 obtained by elongation at various strains for comparison with Figs. 10 and 11.
2)
It is difficult to understand the plasticizing effect of SiO2 mentioned in L268-272. Can the plasticizing effect be explained by the decrease of the glass transition temperature by adding the SiO2? If the stress drop is attributed to the change of the crystalline structure, it is not plasticizing effect.
3)
Figs. 7 and 8 are lacked, so I can not understand the concepts mentioned in L239-251.
4)
To clarify the characteristics of the PA612/SiO2 composites, the authors should cite the data of other composites such as PA6/SiO2 in the literature and compare with the experimental results of Figs. 10 and 11 in this manuscript.
Reviewer 2 Report
In page 7, line 227, is not mentioned if the 2D-WAXD patterns were analyzed under the Fraser corrected procedure. For more detail see: JOURNAL OF POLYMER SCIENCE, PART B: POLYMER PHYSICS 2015, 53, 475–491.
This would improve your study and aport more information of the crystal orientation.
In Page 7, line 240, are missing the Fig. 7 and Fig. 8.
The correlation of mechanical properties under different stretching conditions with the 2D-WAXD patterns must be done, besides of a schematic illustration of the deformation crystal structure.
In page 8, line 291 and 292, the stress-strain curves of stretched PA612 films must be illustrated and discussed .
In page 9, line 303, (Table 3). The glass transition temperature of uniaxially stretched PA612 films must be presented and discussed.
Round 2
Reviewer 1 Report
The previous manuscript has been revised by taking into account the comments. Hence, I would like to recommend this manuscript for publication.
Reviewer 2 Report
Your paper has been enhanced exceptionally. We would expect more studies like this in a future.